# Optimising implementation strategies of the first scaleup of a primary care psychological intervention for common mental disorders in Sub-Saharan Africa: a mixed methods study protocol for the optimised Friendship Bench (OptFB)

Ruth Verhey [1,2] Charmaine Chitiyo,[1,2] Sandra Ngonidzashe Mboweni,[1,2] Ephraim Chiriseri,[2] Dixon Chibanda,[1,2,3] Andy Healey,[4] Bradley Wagenaar,[5,6] Ricardo Araya[4,7]

For numbered affiliations see end of article.

**Correspondence to**
Dr Ricardo Araya;
ricardo.araya@kcl.ac.uk

## ABSTRACT

**Introduction** Common mental disorders (CMDs) are a leading cause of disability globally. CMDs are highly prevalent in Zimbabwe and have been addressed by an evidence-based, task-shifting psychological intervention called the Friendship Bench (FB). The task-shifted FB programme guides clients through problem-solving therapy. It was scaled up across 36 implementation sites in Zimbabwe in 2016.

**Methods and analysis** This study will employ a mixed-method framework. It aims to: (1) use quantitative survey methodologies organised around the Reach, Effectiveness, Adoption and Implementation and Maintenance evaluation framework to assess the current scaleup of the FB intervention and classify 36 clinics according to levels of performance; (2) use qualitative focus group discussions and semistructured interviews organised around the Consolidated Framework for Implementation Research to analyse determinants of implementation success, as well as elucidate heterogeneity in implementation strategies through comparing high-performing and low-performing clinics; and (3) use the results from aims 1 and 2 to develop strategies to optimise the Friendship Bench intervention and apply this model in a cluster randomised controlled trial to evaluate potential improvements among low-performing clinics. The trial will be registered with the Pan African Clinical Trial Registry (www.pactr.org). The planned randomised controlled trial for the third research aim will be registered after completing aims one and two because the intervention is dependent on knowledge generated during these phases.

**Ethics and dissemination** The research protocol received full authorisation from the Medical Research Council of Zimbabwe (MRCZ A/242). It is anticipated that changes in data collection tools and consent forms will take place at all three phases of the study and approval from MRCZ will be sought. All interview partners will be asked for informed consent. The research team will prioritise open-access publications to disseminate research results.

## Strengths and limitations of this study

► Few evidence-based psychological interventions offered at primary healthcare level have been successfully scaled-up in Sub-Saharan Africa; this study is designed to deliver detailed knowledge about factors that influence the scale-up of a primary care psychological intervention (the Friendship Bench) in an African setting.

► Two widely used implementation science models, Reach, Effectiveness, Adoption and Implementation and Maintenance and Consolidated Framework for Implementation Research, will be used to evaluate the implementation of this intervention, which was scaled up in 2016.

► This study focuses on evaluating the scaling up of evidence-based interventions and developing and testing implementation strategies to potentially optimise the routine delivery of the Friendship Bench.

► A limitation is that comprehensive implementation data are only collected three years after the scale up exercise.

## INTRODUCTION

In the past 10 years, it has become apparent that mental, neurological and substance use disorders (MNS) are among the leading causes of the global disease burden.[1–3] Research has shown that 4 out of every 10 people in low-income and middle-income countries (LMICs) suffer from mental disorders (de Boer *et al*, 2008, World Health Organization, 2009a) and evidence-based mental health interventions have become a focus of research and interest.[4] It has been observed that the poor are disproportionately affected by mental disorders.[5 6] Less than 5% of people

living in some LMIC receive any adequate treatment for mental health disorders.[7–9] Particularly in LMIC the lack of resources, especially trained mental health professionals, causes suboptimal detection and management of common mental disorders (CMDs).[10–12] Worldwide, efforts have been made to create sustainable and affordable mental health interventions in primary care.[13–18] In a recent systematic review, only four studies were detected that had evaluated the implementation of a depression intervention scaled up in routine care.[19] As it stands, the benefit of these evidence-based interventions is not yet reaching those populations most at need across LMICs.

Zimbabwe, a country in Southern Africa with a population of 13 million has a large treatment gap for MNS. Studies show that over 30% of primary healthcare (PHC) users need mental healthcare services for mostly CMD and only 5% of these receive appropriate care.[20] Untreated CMD can also lead to worsening of clinical outcomes in chronic conditions such as HIV[21] and negatively affect economic outcomes too.[5] The Friendship Bench (FB) was developed in response to the existing treatment gap for mental healthcare in Zimbabwe and tested for its efficacy in a cluster randomised controlled trial (RCT).[22]

This task-shifted intervention is delivered by trained and supervised lay health workers (LHWs) who deliver problem-solving therapy (PST)[23] on a bench located in PHC clinics. In 2016, the FB intervention was scaled up across Harare, Gweru and Chitungwiza and surrounding peri-urban communities in collaboration with the respective City Health departments.[24] The FB programme was established in 72 City Health PHC clinics that are established in 36 sites (different clinic types can be found in the same site). This scaling-up exercise involved the training of more than 300 LHWs in the 3 cities in Zimbabwe.[24] Maintenance funding for FB activities is provided by the City Health department.

All LHWs working for the FB PHC clinics in Harare, Gweru and Chitungwiza received the standard manualised training and supervision. While existing scientific evidence has shown that under ideal randomised trial conditions the FB intervention leads to clinically significant reductions in symptoms, little implementation research has been carried out regarding the performance of FB under routine conditions as the model is being further scaled up across Zimbabwe.

This study will be of interest to implementation scientists, policy-makers and researchers working to scale-up primary care psychological interventions in LMICs globally. Results from this study have the potential to inform future scaleup and maintenance of task-shared psychological interventions into routine Ministry of Health primary care settings.

## Preliminary observations

Preliminary work had revealed that FB activities were irregular over the implementation sites. FB related data collection was often unreliable due to various reasons such as the delivering agents not having been trained on

data collection, and the FB programme data not being reported to the authorities as part of the clinic activities. Only estimates for client numbers for 2016–2018 with a programme reach decline from 27 967 clients in 2016 to 6688 in 2018 for all of the 36 sites were available. Sites in Harare had continued to offer the programme. In the two other cities (Gweru and Chitungwiza), the health authorities had ceased to support the FB programme and delivering agents had been told to focus on other programmes such as HIV-related activities. It was unclear how many FB activities had been carried out. In order to receive continued support, the FB programme should be integrated with other PHC programmes such as HIV care. Data collection efforts need to be simplified and delivering agents trained. Data need to be gathered and analysed regularly using implementation science principles. Furthermore, the FB organisation should engage closely with healthcare providers and policy-makers to ensure successful and continued programme implementation.

## Overall study goal

This research uses a mixed-method study design and widely used implementation frameworks to systematically analyse the performance of clinics, determinants of this performance, including implementation strategies that might differentiate high-performing versus low-performing clinics, and develop and test an enhanced implementation strategy to improve the performance of clinics in three cities in Zimbabwe. The study is designed to be conducted in three phases with corresponding aims.

First (aim 1), we plan to examine how the FB is performing under real-world implementation conditions and classify existing clinics with FB into high-performing versus low-performing sites using differences in Reach, Effectiveness, Adoption and Implementation and Maintenance (RE-AIM) outcomes.[25 26]

Second (aim 2), we will analyse the determinants of heterogeneity in the results of phase 1 comparing high-performing versus low-performing clinics, mainly using the Consolidated Framework for Implementation Research (CFIR) framework[27] and rigorously documenting changes to the original FB protocol and current implementation strategies in use.

Third (aim 3), we will develop and test an optimised package of FB implementation strategies based on the results of phase 2 and measure the improvement among low performing clinics using RE-AIM outcomes.

## Study setting

The study will be conducted in PHC clinics in Harare, Gweru and Chitungwiza.

Most of the clinics in the three cities are located in comparable areas which are characterised by high population density and informal income generating activities often occurring in the vicinity of the clinics. Depending on their size, PHC clinics serve between 20 000 and 80 000 people from the most socioeconomically disadvantaged sectors of the population. Clinics are differentiated into

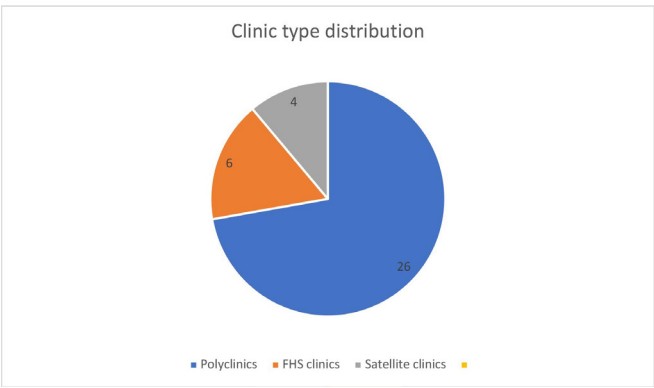

**Figure 1** Clinic type distribution for Harare, Gweru and Chitungwiza. PHC, primary healthcare.

poly, satellite and family health service clinics according to the size of the clinic and the range of services offered.

The most comprehensive services are offered in a Polyclinic such as prenatal, postnatal and perinatal care, opportunistic infections (eg, TB treatment), and specialised NGO-based programmes (HIV testing and management, male circumcision, communicable disease awareness). Satellite and Family health clinics (FHS) offer less services. Medical doctors are not permanently present but hold clinics on specific days in poly clinics. This influences the clinic user population's composition on these particular days (eg, HIV clinic day).

Clinics in Harare, Chitungwiza and Gweru are grouped and located in the same geographical facility and these are counted as one FB implementation site. Data will be collected in 36 implementation sites (n=28 in Harare; n=4 in Gweru; n=4 in Chitungwiza). Of these 26 Poly clinics, 6 are FHS and 4 satellite clinics (see figure 1).

Depending on their size and catchment area, FB implementation sites have between one (1) and fourteen (14) LHWs who deliver the FB intervention on benches in the clinic premises during clinic opening times. Clinic users are informed about the about FB services and mental health through group or individual talks in the clinic's waiting areas. Community members are also directly in contact with LHWs during outreach activities in the community.

## METHODS

This study proposes a rigorous analysis of the multiple interconnecting factors using two internationally recognised implementation research methods—the RE-AIM model[26] and the CFIR[27] which will be described in more detail below. Both conceptual frameworks have been used widely in implementation research for healthcare delivery in order to deepen the understanding and evaluation of interventions such as the FB. The study has three research aims which are linked contextually to each other and are described in detail below.

### Patient and public involvement

Patients and/or the public will be involved in the stakeholder meetings; they were not involved in the design, nor will they be involved in the study conduct, or reporting, or dissemination plans of this research project.

### Methods aim 1

A thorough analysis of the existing routine health information system data collected by the Harare, Gweru and Chitungwiza City Health authorities will be carried out to learn about the FB activities at individual clinic level. These data consist of user numbers, age, gender, HIV status, clients' screening tool scores preintervention and postintervention as well as complete use of screening tool, and number of sessions.

We will use the RE-AIM evaluation framework to evaluate the current implementation performance of the FB intervention after 3 years of implementation experience.

Routinely collected data will be used to assess the FB intervention's real-world and pragmatic performance: *R*each, *E*ffectiveness, *A*doption, *I*mplementation and *M*aintenance. The research team which consists of experienced global mental health researchers and clinicians will develop indicators for each of the RE-AIM domains using the www.re-aim.org website to support us and base our decisions on expert consensus and availability of data. These indicators will then be used to design a questionnaire to guide the RE-AIM related data collection. Each indicator will comprise a numerator and a denominator populated with data collected from the clinic records and the planned observations.

The data on the FB implementation will be analysed for each of the 36 participating clinics. Routinely collected data includes clinical registries for both nurses and LHWs and data from the FB Register (commonly known as the 'green book') where the LHWs record beneficiary information.

In addition, LHWs will be observed during all aspects of their work, including giving health talks, interacting with clients, and delivering the FB intervention. We will observe and record whether all FB related tools such as questionnaires and intervention tools are used.

In order to collect additional necessary data for AIM 1, key respondents will be interviewed using a questionnaire that will be developed by the research team.

We plan to interview at least two LHWs per clinic and in clinics with more than two LHWs; we will interview 50% of the present LHWs by randomly selecting them. Papers with their names will be put in a container from which an RA will pull out the appropriate number in the LHWs' presence. We will always interview the supervisor LHW of each clinic if this position is taken in a particular clinic. We will also interview the nurse in charge in every clinic and the associated district health promoting officers (DHPOs) (n=10). Data will be collected from June to September 2019 in all participating sites.

The data collection will be carried out by two research coordinators who will lead two teams of four trained and

supervised research assistants (RAs). The teams will visit each clinic for 2 days. The clinics will be sensitised about the FB team visit a week prior. The RAs will be trained to interview, to observe and record the FB-related activities in the clinic and how to enter the data digitally using tablet computers. They will be trained on data checking, cleaning and uploading.

Furthermore, we are planning to audio-record FB sessions with consenting clients (two per site, n=72). We will approach, where possible, all incoming clients seeking services and ask them for informed consent to allow us to record their session with the FB LHWs. We aim to record as many as possible but at least two per site.

The recordings will be translated, transcribed and rated according to the FB fidelity checklist.

The FB fidelity checklist assesses for communication skills of the counsellor, the level of psychoeducation that is done, and the adherence to the PST steps that the FB counsellor is trained to deliver (see Supplementary Appendix A for full fidelity checklist which was developed for the RCT[22]). The assessments of audio recordings will be done by trained FB research team members who will prepare an audio recorder which will be left with the FB counsellor after a client has given consent. The audio recording device will be retrieved by the RA when the LHW has indicated that the session is done.

In the event that no clients come to the clinic on both days that the FB team visits the site or no client consents to have their session audio recorded, this will be entered as missing. Due to logistic and financial constraints a repeat visit to a particular clinic will not be possible.

All respondents will be asked to answer the questions with regards to FB activities in the past month. According to their position with regards to FB activities, questions might be formulated slightly differently.

The questionnaires will be administered using tablet computers (Lenovo); all observational data will be entered digitally after their correctness has been ascertained by asking interviewees to show evidence as applicable. Questionnaires and observation guides are programmed into the tablets using Kobotoolbox (https://www.kobotoolbox.org) which is a data collection tool. Collected data will be cleaned and uploaded daily to a password secured server.

The research team will also observe FB-specific activities such as health and 'mobilisation' talks that are given by the clinic staff including the LHWs while patients are waiting to be seen.

A stakeholder meeting will be held once aim 1 data are completed and the data are analysed. At this meeting, the research team will present the results from aim 1 and discuss potential reasons why we might see the differences in implementation across sites with stakeholders. This meeting will be attended by all relevant clinic staff, health authority officers as well as clients. Information from stakeholders will be used to select and prioritise CFIR constructs to include in qualitative interview guides for aim 2.

## Data analysis aim 1

The goal of aim 1 is to classify the 36 FB implementation sites on their performance based on the RE-AIM outcomes. Our methods will follow similar classification efforts previously published.[28] Clinics will be first ranked according to their performance within each individual measure. Clinics score on all indicators within one construct (for example reach) will be averaged. For each of the RE-AIM constructs, every clinic will thus have an averaged ranking.

These domain-based rankings will be averaged per clinic rankings giving an overall ranking by calculating simple means of all domain rankings. This procedure will be carried out by two independent individuals and any differences will lead to a redoing of the process. In case of same outcomes for clinics, we will treat these particular clinics as being on the same rank. This will give us a final composite rank for each clinic which will be used to determine the 10 highest and 10 lowest performing clinics that will be qualitatively assessed in aim 2.

## Methods aim 2

With the aim to understand the determinants of implementation success, as well as differences in implementation strategies employed, aim 2 will use focus-group discussions organised around the CFIR.[29–31] Through these qualitative methods, we aim to gain a deeper understanding of the factors that contribute to the successful implementation comparing high-performing with low-performing clinics. The CFIR framework focuses on an overview of potential multilevel determinants of healthcare delivery. It was designed to help understand integrated implementation determinants across multiple levels (clients; implementers; organisations; contexts; processes).

For the present study, we will focus on determinants of implementation success, taking lessons from both high-performing and low-performing clinics to inform the development of an improved package of implementation strategies targeting identified barriers.

Focus group discussions (FDGs) with key informants (LHWs, nurses, DHPOs, clients) of the 10 high-performing and 10 low-performing clinics will be carried out by trained qualitative researchers. The FB-specific interview guides for these group discussions and interviews will be developed by the study team in a sequence of internal project meetings using the online technical support website wwwcfirguideorg. The results of aim 1 will guide us in designing the interview guides for the FDGs.

The outcome of the stakeholders meeting in which we present the results of aim 1 will also give us insight on the importance of constructs which we will take into account when designing the CFIR interview guides.

Interview guides will be translated into the local language Shona and all group discussions will be audio recorded, transcribed and translated to English. All discussions will be held in the local language.

The FGD participants will be selected from all 10 low-performing and high-performing clinics, respectively. We will interview LHWs, nurses, DHPOs in their role as implementers as well as clients as recipients of the intervention. Nurses and DHPOs will be invited to joined meetings. We will conduct FGDs for all available LHWs at every selected clinic. We will ask the selected LHWs to purposively suggest two clients each, whom we will then invite to FGDs in each of the selected clinics. In case a client declines participation, we will ask for another suggestion.

FDGs will take place in clinics or, if not possible, in the FB office in Harare.

## Data analysis aim 2

CFIR analyses will follow the original Damschroder methodology previously published.[30] Briefly, two independent local Zimbabwean reviewers will code each FGD transcript according to the selected CFIR constructs. Differences will be discussed and revised until final codes are agreed on. Facility-level case memos will be organised by the relevant CFIR construct, using each new transcript to confirm and refine statements until all transcripts are coded. This process will be closely supported by the whole research team. Each clinic will have two case memos, one for LHWs and other implementers and one for clients.

Using case memos and supporting transcripts, the same two coders will independently rate CFIR constructs on valence (X (mixed); 0 (neutral); + (construct has a positive effect on implementation) or − (construct has a negative influence on implementation). Once drafted, the entire research team will meet and use a deliberated consensus to finalise memos, constructs, and valence. These data will be mapped on a matrix template with the goal of identifying constructs that differ between facilities with high and low performance to identify factors relevant for the success of the implementation. Analyses will progress with visual inspection of patterns in constructs and valence by high vs low performing clinics, as well as examining median and mean valence by high-performing versus low-performing clinics. Once distinguishing constructs are identified, the team will re-review case memos and coded transcripts to gather more information on constructs.

## Aim 3

In aim 3, we will develop a package of optimised Friendship Bench (OptFB) implementation strategies matched to key barriers identified in the previous phases of this study. Using CFIR data on barriers/facilitators to high-quality FB implementation, we will use the CFIR-Expert Recommendation for Implementation Change (ERIC) matching tool to examine and select implementation strategies to address key CFIR constructs discriminating between high and low performing clinics in aim 1 (https://cfirguide.org/choosing-strategies/).[32 33] Once a preliminary list is developed by our team, the CFIR-ERIC matching tool[32] will be used to prioritise those strategies that are found to be most likely to address CFIR barriers in low-performing clinics.[33 34]

We will engage in a participatory stakeholder Delphi rating exercise to select specific strategies. This will be followed by the research team specifying and tailoring the strategies for the Zimbabwean context by including the additional information gained from the stakeholders. Aspects of feasibility, affordability and effectiveness will guide this process in order for the package to be meaningful and effective.[35] Strategies currently in use by high-performing clinics will be also considered for the OptFB implementation strategies.

This OptFB package or intervention of improved strategies will be tested in low-performing clinics. Ongoing RE-AIM data are being collected on a monthly basis in each clinic. Using these data on RE-AIM outcomes, we will reclassify clinics using a similar process as in aim 1. We will then identify the 18 lowest performing clinics and randomly select 12 clinics to deliver the OptFB and 6 to act as control clinics over a period of 6 months. The primary outcome will be a composite measure of RE-AIM indicators estimated at 6 months after the commencement of the implementation of the OptFB intervention. We will estimate changes in this composite measure of implementation before and at 6 months after starting the delivery of OptFB in all clinics. We will compare the difference in means or proportions between the clinics receiving the OptFB and the control clinics using the routinely collected data. Secondary outcomes will examine performance of each of the RE-AIM outcomes separately and clinical effectiveness results at individual level. The latter will be based on individual scores on the SSQ on a minimum of 20 random individuals per clinic during the 6-month period.

No sample size calculation has been estimated since there are no previous studies on which to estimate an effect size, the number of clinics is small, and the main outcomes are averaged data representing clusters. Nonetheless, we expect to see larger improvements in the RE-AIM composite index score in the clinics receiving OptFB compared with the control clinics over the 6 months. As a secondary outcome measure, clinical effectiveness will be assessed based on changes on SSQ scores from baseline to 6 months for a sample of 360 individuals (18 clinics with 20 individuals each), but we do not expect this sample would have enough power to detect small differences in effectiveness across the two group of clinics. Thus, comparisons on clinical effectiveness must be considered purely descriptive and exploratory and interpreted with caution. In any case, the main outcomes of interest in this study are implementation outcomes subsumed under the domains included in the RE-AIM framework.

## Data analysis aim 3

We will use a difference-in-differences analysis comparing the groups over time. Means or proportions on outcome data will be compared across groups using descriptive

statistics. Regression models will be used to estimate the effect of the intervention on the main outcomes. General estimating equations with robust standard errors will be used to control for clustering. Potential confounders will be determined a priori and included in the regression models. Standard errors, confidence intervals and p values will be obtained. A similar secondary analysis will be conducted with the secondary outcome measures.

## Health economic analysis

Site-level data will be collected on fidelity to the OptFB implementation strategies, along with activities and resource inputs required to deliver improvement strategies and OptFB delivery costs. Economic modelling will be used to combine this information with data and evidence on clinical impact and implementation effectiveness to evaluate the cost effectiveness of the OptFB programme.[36]

We will also revisit clinics and re-engage with stakeholders in FGD to explore level of change in the identified CFIR domains in the intervention arm clinics.

After completion of the trial, the strategy will also be implemented in the control arm clinics to increase the overall performance in all of participating lower performing clinics.

## DISCUSSION

This study will contribute to the knowledge about scaling up of an evidence-based task-shifted intervention in a LMIC. This is a unique opportunity to analyse the FB in a real-world setting. As mentioned above, not many interventions have been scaled up from LMICs and therefore there is a dearth of information on how implementation strategies can be used in order to ensure a strong scaling up. With this study, we hope to learn which barriers and enablers are at play in the FB scale up process. This is particularly important for us as we are expanding the FB services throughout Zimbabwe and beyond to meet the population's needs for accessible and acceptable mental healthcare. This effort has to be undertaken with the aim of having high fidelity to the programme while considering contextual aspects. Using implementation science principles will help us to give theoretical justification and describe specifications for application for those implementation strategies that we will devise after having gone through the different stages of this research process. Evidence-based, clear and applicable guidelines of how to implement our evidence-based intervention in PHC settings will be created and can then subsequently be used to ensure a strong implementation of FB.

## Ethics and dissemination

This research protocol has been approved by the Medical Research Council Zimbabwe (MRCZ), MRCZ/A/2428 and the Joint Research Council (JREC), 79/19. Results will be disseminated in peer-reviewed journals and conferences.

**Author affiliations**
[1]Research Support Centre, University of Zimbabwe, Harare, Harare, Zimbabwe
[2]Friendship Bench Zimbabwe, Harare, Zimbabwe
[3]Global Mental Health, London School of Hygiene and Tropical Medicine, LSHTM, London, UK
[4]IOPPN, King's College London, London, UK
[5]Department of Epidemiology, University of Washington, Seattle, Washington, USA
[6]Department of global health, University of Washington, Seattle, Washington, USA
[7]Centre for Global Mental Health and Primary Care Research, London, UK

**Contributors** RA designed the study and received the grant. RA is the overall PI. RV and DC are leading the study locally. CC will have oversight as project coordinator together with SNM as her assistant over the data collection in all phases. BW is leading on the implantation science aspects. EC is conducting data cleaning and assisting with analysis. AH is leading on the health economics analysis. All authors will contribute to the development of questionnaires, interview guides and the strategies for the intervention. RV wrote the first version of the protocol paper. All authors contributed by critically reviewing all further drafts and approving of the final paper.

**Funding** This work was supported by Global Alliance for Chronic Diseases (GACD) through the Medical Research Council Grant number: MRC UKRI MR/S004270/1.

**Competing interests** None declared.

**Patient consent for publication** Not required.

**Provenance and peer review** Not commissioned; externally peer reviewed.

**ORCID iD**
Ruth Verhey http://orcid.org/0000-0002-5959-1891

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
