## [Reviewer comments · BMJ Open]

ARTICLE DETAILS

TITLE (PROVISIONAL)	Optimizing implementation strategies of the first scale-up of a primary care psychological intervention for common mental disorders in Sub-Saharan Africa: A Mixed Methods Study protocol for the Optimized Friendship Bench (OptFB)
AUTHORS	Verhey, Ruth; Chitiyo, Charmaine; Mboweni, Sandra; Chiriseri, Ephraim; Chibanda, Dixon; Healey, Andy; Wagenaar, Bradley; Araya, Ricardo

VERSION 1 – REVIEW

REVIEWER	Bhana, Arvin University of KwaZulu-Natal College of Health Sciences, School of Nursing and Public Health
REVIEW RETURNED	03-Dec-2020

GENERAL COMMENTS	1. Minor typological errors: Page 16: Line 24 "into should read "in".2. To what extent is the FB integrated into mental health care treatment part of health service provision offered by the public health facilities? The description of the FB in Zimbabwe indicates that it is funded by the City Health Department. Does this mean that it is integrated as the authors refer to expanding FB services throughout Zimbabwe! This should be clarified as the FB intervention needs to be interpreted correctly as being outside of the primary health care system providing a mental health care service to the Zimbabwean health care system.2. To maximise the benefits of using RE-AIM and CFIR frameworks, it is important to know if the study sites differ from each other. Reference is made to most of the sample clinics offering comprehensive services (26 polyclinics), while 10 clinics offer a more limited service. They also differ in relation to the number of lay health workers attached to these clinics. What is the specific research objective in including clinics with different staff complements and size? How will high and low performing clinics be understood in relation to these differences?3. LHWs constitute a key element in an understanding of real-world implementation. It is important to provide a more comprehensive description of the implementers. LHWs may all receive similar training but may differ in relation to their relative experience in working with either FB or other mental health interventions. For example, how many LHWs are also trained in HIV counselling?4. Could the authors clarify why users experience of the FB is not part of the RE-AIM evaluation (I am aware of the cluster RCT showing efficacy)?
--

	Page 8: Lines 55-58 - meaning of the sentence is unclear Page 11: Lines 8-12 - what is the frequency of these observations? Is it when data is being collected for RE-AIM and CFIR? Page 12: Lines 36-37: The procedure describes the ranking of clinics based on their performance within each RE-AIM dimension. I would suggest that these rankings should be evaluated by at least two individuals independently and reference disagreements with an indication of how these differences were resolved to ensure that the entire process is robust.
--	---

REVIEWER	Bobevski, Irene Monash University, Department of Psychiatry, Monash University
REVIEW RETURNED	22-Jan-2021

GENERAL COMMENTS	This protocol outlines an important and interesting study. Some clarifications are required, as per my comments below. Methods Aim 1 For Aim 1, data will be examined "after three years of implementation experience": is there information available on how many clients have participated in the FB intervention during this time? How many clients would have participated in the FB program per week/month for each clinic? The data for Aim 1 includes clients' screening tool scores. Does this include outcome data post-intervention or just pre-intervention screening? Are there any existing validated indicators for the RE-AIM domains which could potentially be used or adapted? For the observation of the LHWs for Aim 1, will all LHWs will be observed or will a sample be selected (and if so how will it be selected)? Regarding the recording of sessions for 72 clients (p. 8, l.22), how will the 72 clients be selected? Methods Aim 2 Will focus groups be carried out separately for each clinic? How will focus group participants be selected? Aim 3 It is surprising that clinical effectiveness is not included as a primary outcome of the OptFB trial (p.14, l.26). Unless the program is clinically effective there is no benefit in implementing it. An earlier RCT study (reference 22) has shown FB to be clinically effective at the individual level, however it is important to evaluate this in the scaled-up OptFB as a primary outcome. Even if the study has no power to detect small differences in clinical effectiveness among the clinics (p.14, l.37-47), it is still important to evaluate at least the overall clinical effectiveness of the OptFB. During the 6 months period of the RCT how many clients in total would be expected to participate in the FB program?
---

	The authors state that "No sample size calculation has been estimated since there are no previous studies on which to estimate the effect size, the number of clinics is small, and the main outcomes are averaged data representing clusters." (p.14, l.29-33). It may be possible to hypothesise effect sizes based on results from Aims 1 and 2, and to present some sample size estimates controlled for clustering (perhaps for at least detecting a medium effect size). For the clinical effectiveness outcome, effect sizes can be obtained from the previous RCT described in reference 22.
--	---

VERSION 1 – AUTHOR RESPONSE

Reviewer: 1

Prof. Arvin Bhana, University of KwaZulu-Natal College of Health Sciences

Comments to the Author:

1. Minor typological errors:

Page 16: Line 24 "into should read "in".

Thank you for finding this error. We have replaced “into” with “in” in the manuscript.

2. To what extent is the FB integrated into mental health care treatment part of health service provision offered by the public health facilities? The description of the FB in Zimbabwe indicates that it is funded by the City Health Department. Does this mean that it is integrated as the authors refer to expanding FB services throughout Zimbabwe! This should be clarified as the FB intervention needs to be interpreted correctly as being outside of the primary health care system providing a mental health care service to the Zimbabwean health care system.

Dear reviewer, thank you for this important question. The FB is part of the Ministry of Health’s care plan for Zimbabwe as described in a memorandum of understanding between the two entities. The FB services are integrated into the primary health care system in Zimbabwe, albeit not offered yet in all primary health care clinics all over the country. Currently, FB services are found in 36 sites in 3 cities. For this study, we focus on the described 36 sites.

2. To maximise the benefits of using RE-AIM and CFIR frameworks, it is important to know if the study sites differ from each other. Reference is made to most of the sample clinics offering comprehensive services (26 polyclinics), while 10 clinics offer a more limited service. They also differ in relation to the number of lay health workers attached to these clinics. What is the specific research objective in including clinics with different staff complements and size? How will high and low performing clinics be understood in relation to these differences?

Thank you for this insightful comment. Indeed, the different sizes of clinics influence the range of services. Nevertheless, the delivering agents who are employed as health promoters by the city health authorities work somewhat parallel to the clinic staff that are trained medically (maternity and childcare, infectious and non-infectious disease care). As the lowest ranking workforce and often busy with outreach programs, they are overseen by a health promoting officers who is responsible for all activities within a district. We are aware that smaller clinics might not have a peer supervisor for the

FB services and therefore less immediate support. From the scale up perspective, the delivering agents have been trained in the same way and were given the same Friendship Bench work tools. We hope to understand which factors do influence the service delivery for all involved clinics irrespective of their size.

3. LHWs constitute a key element in an understanding of real-world implementation. It is important to provide a more comprehensive description of the implementers. LHWs may all receive similar training but may differ in relation to their relative experience in working with either FB or other mental health interventions. For example, how many LHWs are also trained in HIV counselling?

We thank the reviewer for this comment that is so important. We believe that community or lay health workers are so often being undervalued in their experience, expertise and contribution to primary care. We hope to understand more about their perspective by using the CFIR framework to guide us once we have classified the clinics. As there was no primary care based mental health intervention before the Friendship Bench (FB), the specific group of FB service delivering agents had not received mh related training before the FB scale up exercise. HIV counselling is offered by a different group in the PHC system in Zimbabwe. They are called primary care counsellors. In future, we do foresee that the FB intervention will be offered by primary care counsellors to, especially as we realize the importance of supporting the HIV care efforts to include evidence-based mh components.

4. Could the authors clarify why users experience of the FB is not part of the RE-AIM evaluation (I am aware of the cluster RCT showing efficacy)?

We really appreciate your comment! We plan to make use of the RE-AIM model to guide us to classify the clinics based on their performance and to collect mostly quantitative data that explore the quality of the suggested domains. At this point, we plan to involve service users in aim 2 to explore barriers and enablers for the service implementation. In future, our research group will make sure that we involve service users in all phases.

Page 8: Lines 55-58 - meaning of the sentence is unclear

Unfortunately, we were not able to detect the sentence that you are referring to.

Page 11: Lines 8-12 - what is the frequency of these observations? Is it when data is being collected for RE-AIM and CFIR?

Thank you for this question. We plan to carry out relevant observations when we visit each clinic for the aim data collection for two consecutive days. We understand that it is a limitation of our study that we will only have a restricted number of observations per site.

Page 12: Lines 36-37: The procedure describes the ranking of clinics based on their performance within each RE-AIM dimension. I would suggest that these rankings should be evaluated by at least two individuals independently and reference disagreements with an indication of how these differences were resolved to ensure that the entire process is robust.

Dear reviewer, thank you for pointing this out. We will take your suggestion into account to ensure a robust process. We have changed the text in chapter 4.1.1 Data Analysis Aim 1 and added the following sentence on page 9:

This procedure will be carried out by two independent individuals and any differences will lead to a redoing of the process.

Reviewer: 2

Dr. Irene Bobevski, Monash University

Comments to the Author:

This protocol outlines an important and interesting study. Some clarifications are required, as per my comments below.

Methods Aim 1

1. For Aim 1, data will be examined "after three years of implementation experience": is there information available on how many clients have participated in the FB intervention during this time? How many clients would have participated in the FB program per week/month for each clinic?

Dear reviewer, thank you for this important point. Preliminary work has revealed that record keeping in the clinics was often unreliable due to various reasons such as the delivering agents not having been trained on data collection, and the FB program data not being reported to the authorities as part of the clinic activities. Therefore, we only have estimates of client numbers for 2016-2018 with program reach decline from 27,967 clients in 2016 to 6,688 in 2018 for all the 36 sites.

2. The data for Aim 1 includes clients' screening tool scores. Does this include outcome data post-intervention or just pre-intervention screening?

Thank you for this comment. While we are interested in pre- and post intervention scores to learn about the effectiveness of the program during the scale up period, we are also interested in whether screening tools have been used reliably.

We have changed the sentence, "pre- and post-intervention as well as complete use of screening tool", and it now reads like this:

This data consists of user numbers, age, gender, HIV status, clients' screening tool scores pre- and post-intervention as well as complete use of screening tool, and number of sessions.

Are there any existing validated indicators for the RE-AIM domains which could potentially be used or adapted?

Thank you for this thoughtful comment: Our research team intends to use the RE-AIM website to help us design indicators that will be specific to our setting. We have changed the sentence and added:

"using the www.re-aim.org website to support us and base our decisions". It now reads as follows:

The research team which consists of experienced global mental health researchers and clinicians will develop indicators for each of the RE-AIM domains using the www.re-aim.org website to support us and base our decisions on expert consensus and availability of data.

For the observation of the LHWs for Aim 1, will all LHWs will be observed or will a sample be selected (and if so how will it be selected)?

We thank the reviewer for this comment. Depending on clinic sizes, we intend to observe/interview a sample of half of the LHWs of each implementation site. We will select them randomly by choosing amongst the ones who are present on the observation day through pulling their names from a container.

Please would the reviewer tell us whether the description of the process on page 7, row 27 onwards is sufficient?

Regarding the recording of sessions for 72 clients (p. 8, l.22), how will the 72 clients be selected?

Thank you, dear reviewer, for this important question! During the two days that the data collection team will visit a site, we will approach all clients who seek services and ask for their informed consent to record the session. We hope to get as many recordings as possible but will seek to get at least 2 per site. We will thus not be able to randomize the client selection but rather ask all clients. We know that this number is very small compared to the possible number of LHWs in a bigger site and will not be representative and this is a limitation to our study.

We have added the following sentences:

We will approach, where possible, all incoming clients seeking services and ask them for informed consent to allow us to record their session with the FB LHWs. We aim to record as many as possible but at least two per site.

Methods Aim 2

Will focus groups be carried out separately for each clinic? How will focus group participants be selected?

Thank you for this comment which points out how important it is to explain the process in detail.

The high performing clinics' participants will be in separate FGDs from the low performing clinics' participants in order for us to learn unbiased information.

All nurses and DHPOs responsible for the selected clinics will be invited together based on their clinic's classification result in their respective cities. We will separate the LHWs from their superiors as we have noticed that hierarchical job structures can hinder the discussion process for some.

We will invite the LHWs of each of the clinics in both segments. We will ask the selected LHWs to purposively suggest 2 clients each, whom we will then invite to FGDs in each of the selected clinics. In case a client declines participation, we will ask for another suggestion. Participants from the two other cities (Gweru and Chitungwiza) will be invited to travel to Harare for the FGD.

We have added the following sentences to the manuscript on page 10:

Nurses and DHPOs will be invited to joined meetings. We will conduct FGDs for all available LHWs at every selected clinic. We will ask the selected LHWs to purposively suggest 2 clients each, whom we will then invite to FGDs in each of the selected clinics. In case a client declines participation, we will ask for another suggestion.

Aim 3

It is surprising that clinical effectiveness is not included as a primary outcome of the OptFB trial (p.14, l.26). Unless the program is clinically effective there is no benefit in implementing it. An earlier RCT study (reference 22) has shown FB to be clinically effective at the individual level, however it is important to evaluate this in the scaled-up OptFB as a primary outcome. Even if the study has no power to detect small differences in clinical effectiveness among the clinics (p.14, l.37-47), it is still important to evaluate at least the overall clinical effectiveness of the OptFB.

During the 6 months period of the RCT how many clients in total would be expected to participate in the FB program?

The authors state that "No sample size calculation has been estimated since there are no previous studies on which to estimate the effect size, the number of clinics is small, and the main outcomes are averaged data representing clusters." (p.14, l.29-33). It may be possible to hypothesise effect sizes based on results from Aims 1 and 2, and to present some sample size estimates controlled for clustering (perhaps for at least detecting a medium effect size). For the clinical effectiveness outcome, effect sizes can be obtained from the previous RCT described in reference 22.

Thank you for this very relevant comment. We expect to learn precise details about the reach of the program from the aim 1 results. Formative research has shown that the data collection in the sites has been erratic and there is a lack of reliable data. Due to the way clinical data was recorded previously, we are unable to link patient clinical outcomes over time across a client's visits to see a change in symptoms. We are trying to improve the data linking process to include these outcomes in future, it is not feasible to do so at the present time. Ideally, we expect to see up to 4 clients per week per LHW based on 180 active delivering agents (n=2880 clients/week).

General correction:

Page 12, row 14-15: Deleted the word "to" and replaced it with "on the". The sentence now reads as follows:

The latter will be based on individual scores ~~to~~ on the SSQ on a minimum of 20 random individuals per clinic during the 6-month period.

Reviewer: 1

Competing interests of Reviewer: None declared

Reviewer: 2

Competing interests of Reviewer: None declared

VERSION 2 – REVIEW

REVIEWER	Bobevski, Irene Monash University, Department of Psychiatry, Monash University
REVIEW RETURNED	13-Apr-2021

GENERAL COMMENTS	1."Preliminary work has revealed that record keeping in the clinics was often unreliable due to various reasons such as the delivering agents not having been trained on data collection, and the FB program data not being reported to the authorities as part of the clinic activities. Therefore, we only have estimates for client numbers for 2016-2018 with program reach decline from 27,967 clients in 2016 to 6,688 in 2018 for all of the 36 sites." The above needs to be stated in the protocol, including any explanations that the authors may have for the decline, as well as how record keeping reliability would be improved in the future for
--

	the study and the FB to be able to progress. Was the FB program deliberately scaled down or are there other reasons? "We expect to learn precise details about the reach of the program from aim 1 results. Formative research has shown that the data collection in the sites has been erratic and there is a lack of reliable data. Due to the way clinical data was recorded previously, we are unable to link patient clinical outcomes over time across a client's visits to see a change in symptoms. We are trying to improve the data linking process to include these outcomes in future. It is not feasible to do so at the present time. Ideally, we expect to see up to 4 clients per week per LHW based on 180 active delivering agents (n=2880 clients/week)." Again, the above needs to be stated in the protocol, along with how it is planned to address these shortcomings. My initial comment remains, that unless the program is shown to be clinically effective there is no benefit in implementing it. At this point it seems that it is not possible to establish clinical effectiveness due to unreliability in implementation and data collection. This raises questions of whether it is feasible to adequately implement and study the FB program. It is imperative that the authors address these issues in the protocol.
--	---

VERSION 2 – AUTHOR RESPONSE

Comment 1:

"Preliminary work has revealed that record keeping in the clinics was often unreliable due to various reasons such as the delivering agents not having been trained on data collection, and the FB program data not being reported to the authorities as part of the clinic activities. Therefore, we only have estimates for client numbers for 2016-2018 with program reach decline from 27,967 clients in 2016 to 6,688 in 2018 for all of the 36 sites."

The above needs to be stated in the protocol, including any explanations that the authors may have for the decline, as well as how record keeping reliability would be improved in the future for the study and the FB to be able to progress. Was the FB program deliberately scaled down or are there other reasons?

Dear reviewer! Thank you for pointing out that the information should be stated in the manuscript. I have changed the manuscript and added the following paragraph on page 5 under the subheading "preliminary observations":

Preliminary observations

Preliminary work had revealed that FB activities were irregular over the implementation sites. FB related data collection was often unreliable due to various reasons such as the delivering agents not having been trained on data collection, and the FB program data not being reported to the authorities as part of the clinic activities. Only estimates for client numbers for 2016-2018 with a program reach

decline from 27,967 clients in 2016 to 6,688 in 2018 for all 36 sites were available. Sites in Harare had continued to offer the program. In the two other cities (Gweru and Chitungwiza) the health authorities had ceased to support the FB program and delivering agents had been told to focus on other programs such as HIV related activities. It was unclear how many FB activities had been carried out. To be supported continuously by the health authorities, the FB program should be integrated with other PHC programs such as HIV care. Data collection efforts need to be simplified and delivering agents trained. Data needs to be gathered and analysed regularly using implementation science principles. Furthermore, the FB organization should engage closely with health care providers and policy makers to ensure successful and continued program implementation.

Comment 2:

"We expect to learn precise details about the reach of the program from aim 1 results. Formative research has shown that the data collection in the sites has been erratic and there is a lack of reliable data. Due to the way clinical data was recorded previously, we are unable to link patient clinical outcomes over time across a client's visits to see a change in symptoms. We are trying to improve the data linking process to include these outcomes in future. It is not feasible to do so at the present time. Ideally, we expect to see up to 4 clients per week per LHW based on 180 active delivering agents (n=2880 clients/week)."

Again, the above needs to be stated in the protocol, along with how it is planned to address these shortcomings. My initial comment remains, that unless the program is shown to be clinically effective there is no benefit in implementing it. At this point it seems that it is not possible to establish clinical effectiveness due to unreliability in implementation and data collection. This raises questions of whether it is feasible to adequately implement and study the FB program. It is imperative that the authors address these issues in the protocol.

Dear reviewer, thank you for this valuable comment.

I agree fully with you that a program such as the Friendship Bench must show its clinical effectiveness. We have carried out a successful randomized controlled trial in 2015 (Chibanda, Dixon, et al. "Effect of a primary care-based psychological intervention on symptoms of common mental disorders in Zimbabwe: a randomized clinical trial." *Jama* 316.24 (2016): 2618-2626).

Furthermore, we hope to learn from this study how to identify the aspects that have been and will be found to be barriers to the implementation and improve them. With the newly gained knowledge, we will be able to evaluate the program on many domains such as conceptualized in the RE-AIM model for example. It is our goal to be able to carry out replications of the Friendship Bench model with a sound scientific underpinning, and we can learn from the experience of the first scale up exercise in Zimbabwe.